# The Effectiveness of Exercise on Cognitive Performance in Individuals with Known Vascular Disease: A Systematic Review

**DOI:** 10.3390/jcm8030294

**Published:** 2019-03-01

**Authors:** Alyssa Brunt, David Albines, Diana Hopkins-Rosseel

**Affiliations:** 1School of Rehabilitation Therapy, Faculty of Health Sciences, Queen’s University, Kingston, Ontario, ON K7L 3N6, Canada; alyssa.brunt@queensu.ca or alyssa.brunt@gmail.com (A.B.); d.albines@queensu.ca or davidalbines@gmail.com (D.A.); 2Research Appointment, Kingston Health Sciences Centre, Hotel Dieu Hospital, 166 Brock Street, Kingston, Ontario, ON K7L 5G2, Canada

**Keywords:** cardiovascular disease, heart disease, vascular disease, stroke, transient ischemic attack, exercise, exercise movement technique, sports, cognition, awareness

## Abstract

Patients with known vascular disease are at increased risk for cognitive impairments. Exercise has been shown to improve cognition in healthy elderly populations and those with mild cognitive impairments. We explored the literature to understand exercise as a modality to improve cognition in those with vascular disease, focusing on dose-responses. A systematic review was conducted through 2017 using Cumulative Index to Nursing and Allied Health Literature (CINAHL), Cochrane, Ovid Embase, and Ovid MEDLINE databases. Eligible studies examined effects of exercise on memory and cognition in cardiovascular (CVD) or cerebrovascular disease (CBVD). Data extracted included group characteristics, exercise dosage and outcomes measures employed. Twenty-two studies (12 CVD, 10 CBVD) met the inclusion criteria. Interventions included aerobic, resistance, or mixed training, with neuropsychological test batteries assessing cognition. In CVD populations, five studies demonstrated improved cardiovascular fitness and cognition with aerobic training, and another seven studies suggested a dose-response. In CBVD trials, four studies reported improved cognition, with no effects observed in the fifth study. Another study found enhanced cognition with resistance training and four demonstrated a positive association between functional capacity and cognition following combined aerobic and resistance training. Exercise is able to positively affect cognitive performance in those with known vascular disease. There is evidence to suggest a dose–response relationship. Further research is required to optimize prescription.

## 1. Introduction

According to the World Health Organization, cardiovascular disease (CVD) is the foremost noncommunicable disease (NCD) worldwide, contributing to 17 million deaths (48% of NCD deaths) annually and 10% of the global disease burden [1]. This, in turn, results in a high economic burden [2]. By 2030, it is predicted that 23 million people per year will die from CVD [2]. The primary load is largely attributed to either ischemic heart disease or cerebrovascular disease [3]. Along with the more evident health issues which present with CVD, there is evidence indicating individuals with CVD may be at risk for deteriorating cognition, including memory functions, which is independent of age-related decline [4]. Cerebrovascular disease (CBVD) is another source of high morbidity and mortality across Europe and globally, with the prevalence increasing in some younger adult populations [5,6]. Interestingly, the prevalence of CBVD is far greater than that of symptomatic stroke [6]. It has been reported that, for every symptomatic stroke, there are approximately 10 silent brain infarcts [7]. With these alarming statistics, there is a call to better understand this pathophysiology and, more importantly, to find the most appropriate interventions for preventing, delaying or mitigating cognitive decline in those with CVD and CBVD. To provide a context for our review, we explored the literature for areas of research shedding light on the links between CVD and CBVD and cognitive decline.

### 1.1. Atherosclerosis

Intima-media thickness (IMT) is a clinical measure to determine arterial wall changes seen in vascular diseases [8]. Clinically, an IMT value greater than 0.9 mm indicates atherosclerotic vascular disease [8]. Haley et al. (2007) questioned if there was a relationship with IMT and cognitive performance [9]. To answer this question, they recruited CVD patients and created two groups according to high IMT (>0.9 mm) and low IMT (<0.9 mm) measured from the common carotid artery. A steeper decline in cognitive performance was found with the high IMT group. It was concluded that a correlation exists between increases in IMT and significantly poorer performance on neuropsychological tests that assess attention-executive-psychomotor domains [9]. This finding remained true in a subsequent 2-year study whereby Sander and colleagues (2010) followed a large group of subjects (*n* = 3386) over the 2-year period, with their main outcomes being cognitive function using the Blessed Information Memory Concentration Scale (6CIT) and IMT measured from the common carotid artery [10]. The researchers created two groups, those with an IMT ≤1.00 mm (L-IMT) and those with an IMT >1.00 mm (H-IMT). The H-IMT group had poorer performance on the 6CIT at baseline and at the follow up, but also showed a significantly greater decline in performance when compared to L-IMT. Interestingly, they noticed that data showed that participants who were not classified as having cognitive impairments at baseline, but developed impairments 2 years later, were less physically active during the study.

### 1.2. Hypertension

Hypertension is known to be associated with CVD, increased risk for stroke, and vascular dementia (VaD) [11,12,13,14,15]. Research has shown that those with hypertension are more likely to have white matter lesions, or white matter hypertensities (WMH), compared to those with normal blood pressure [15]. Increases in WMH has been found to affect performance on neuropsychological tests, specifically in attention and psychomotor speed [12]. To better understand this correlation, Gunstad et al. (2005) studied this relationship in participants who had been diagnosed with CVD. Using neuroimaging, they found that a lack of systolic blood pressure variability was best able to predict total WMH [16]. Authors discussed that this relationship may imply reduced autoregulation as a possible cause for their results. This conclusion was validated by Keary et al. (2007), whose findings showed better cognitive performance in those with greater blood pressure variability [17]. They rationalized that greater blood pressure variability is an indicator of a vascular system that can compensate for poor cardiac output (CO) and other CVD effects [17]. In addition to WMH, untreated hypertension has been shown to increase the stiffness of arterial walls and reduce their elasticity, which is most detrimental to cerebral small vessels, as it exposes arteries to higher pressures [17,18]. If hypertension is untreated, there is evidence of increased white matter lesions, lacunar infarcts, cerebral microbleeds, and a decline in cerebral blood flow [18,19]. Lastly, hypertension decreases grey matter volume, reducing cognition over time, and has been associated with increased deposition of amyloid plaques (a risk marker of Alzheimer’s Disease (AD)) [19]. Both an increase in silent infarcts and WMHs are strongly linked to future stroke, with an up to 2- to 3-fold increase in risk [6].

### 1.3. Cardiac Output

The function of autoregulation is important as it ensures that an adequate volume of blood is directed to the brain, in preference to other tissues [20]. However, autoregulation mechanisms may be less effective when there is a chronic reduction in systemic blood flow [20,21]. Cardiac output is used to determine systemic perfusion [21]. Jefferson et al. (2007) led a study to determine if participants with lower CO values, or systemic hypoperfusion, would result in lower executive function [21]. A group of community-dwelling participants (mean age 69.14 years) with a history of CVD were divided into two groups, those with low and high CO, respectively. They found that cognitive assessments demonstrated a significant difference between groups, with the high CO group out-performing the low CO group. However, there was no significant decline in memory due to low CO. Their conclusion: systemic hypoperfusion is associated with executive function decline. In addition, the authors suggested that subcortical structures may be more vulnerable to hypoperfusion. These conclusions were corroborated by a magnetic resonance imaging (MRI) study where researchers reported an inverse association between systemic blood flow and WMH in subcortical nuclei, independent of age [20]. Damage in subcortical nuclei would disrupt the frontal-subcortical circuitry required for executive functioning [20].

### 1.4. Cerebral Blood Flow

A gradual decline of cerebral blood flow (CBF) is considered a normal part of aging [22]. Further, it is believed that a decline in CBF is related to declining cognitive function. Lucas and colleagues (2011) compared healthy young (average age 24) and healthy older (average age 63) individuals using a Stroop task as their measure of cognitive performance [23]. Within this study, CBF was quantified by blood flow velocity in the right middle cerebral artery (MCA). At rest, healthy older individuals had a 32% reduction in CBF, slower response time, and fewer correct responses, compared to healthy young individuals. During the exercise protocol, participants completed the Stroop task; as expected, the healthy younger group consistently outperformed the older group. Cerebral hypoperfusion is common among those who have Alzheimer’s Disease (AD), VaD, and in patients with mild cognitive impairment (MCI) [24]. In addition, evidence indicates that cerebral hypoperfusion is one of the initial mechanisms which leads to cognitive impairment [24].

### 1.5. Aerobic Capacity

Starting from the age of 30, there is a gradual decrease in brain structural integrity over time [25,26,27,28]. However, aerobic fitness has been found to preserve grey and white matter integrity [29]. A study recruited healthy older participants (ages 55 to 79) and assessed their maximal oxygen uptake (VO2max), a measure of aerobic fitness, followed by MRI [29]. They found that participants with lower VO2max showed age-related declines in the grey matter density, specifically in prefrontal, superior parietal, and middle/inferior temporal cortices, as well as in the anterior white matter tracts. However, participants with higher VO2max levels showed a sparing effect in those same areas.

### 1.6. Exercise as an Intervention

Colcombe et al. (2006) decided to investigate if a 6-month aerobic exercise intervention could improve brain structural integrity [30]. Using a randomly controlled trial (RCT) study design, investigators had study participants (ages 60–79 years) participate in either aerobic training (intensity starting at 40–50% heart rate reserve (HRR) or whole-body stretching for 1 hour, 3 times per week. Study participants did not differ in VO2max at baseline. At the end of the 6-month intervention, those in the aerobic training group had a significant improvement in VO2max, as expected. Surprisingly, MRI data also demonstrated a significant difference in pre- and post-intervention brain volume preservation for the aerobic training group. The largest changes in grey matter were found in the frontal lobe and the largest changes in the white matter were found in the anterior white matter tracts.

Smith et al. (2010) undertook a meta-analysis of published RCTs looking at exercise and associated interactions on neurocognitive performance, in healthy older adults (between January 1966 and July 2009) [31]. Intervention duration ranged from 6 weeks to 18 months, with brisk walking or jogging being the most common exercise modality [31]. Overall, they reported that aerobic exercise results in modest improvements in attention and processing speed, executive function, and memory, with no observed benefits in working memory. Further, the authors noted that aerobic and strength training had an additive effect for attention, processing speeds, and working memory when compared to aerobic exercise alone. The authors reflected that a potential rationale for seeing greater improvements of the combined interventions may have been more a result of the greater reduction in the numbers of cerebrovascular risk factors paired with the improving aerobic fitness, than just an improved aerobic capacity on its own. This was supported by a subsequent study which concluded that, within the CVD population, additional benefits were seen when combining exercise interventions to include both aerobic and resistance training when looking at peak VO2 [32]. Lastly, Smith et al. (2010) reported that longer interventions showed further improvements in attention and processing speeds [31].

### 1.7. The Research Question

As mentioned, there are various factors that have been linked to poor cognition, some of which are a natural process of aging. However, there appears to be an accelerating or exacerbating effect in those who have CVD or CBVD. Fortunately, prior findings relating to physical activity (PA) and exercise have indicated positive effects in healthy older adults. However, there has been limited research investigating whether these cognitive benefits are expected specifically in individuals with known vascular disease. Therefore, we undertook an updated systematic review of the literature to determine the level of support for this link and, more specifically, to answer the question of the dosage of exercise needed to affect cognition, including memory functions, in individuals over the age of 50 with known CVD or CBVD.

## 2. Methods

### 2.1. Study Types, Participants, and Outcomes

Studies that examined the effect of exercise interventions on memory and cognition in participants of any age with known vascular disease, including HF, atherosclerosis, peripheral vascular disease, and a history of stroke due to vascular atherosclerosis, were eligible for inclusion in this review. Those with non-ischemic cardiomyopathy, without known cerebrovascular accident or hemorrhagic insult were excluded from this review. Because it is a prevalent co-morbidity in populations with vascular disease, studies employing participants with metabolic disorders such as Diabetes Mellitus were not excluded. On the other hand, investigations involving transplant populations and participants with CVD coinciding with pulmonary disease or renal disease were excluded. Non-randomized and case-control studies were also eligible, as the number of relevant RCTs investigating this topic was predicted to be low, especially within the CVD population. Papers not published in English and those involving animal participants were excluded from the review. To promote a focus on current work, studies published earlier than 1995 were also excluded from this review.

Exercise interventions were defined as any mode of aerobic or resistance activity including, but not limited to, those prescribed within cardiac rehabilitation (CR) programs, as well as those related to functional, leisure, and sports physical activities. To avoid prematurely limiting the findings around this topic, no exclusion criteria on length of intervention or follow-up period was imposed. Studies involving any type of cognitive training in either the control or intervention group were excluded from the search to avoid confounding our findings. However, studies involving interventions which compared the effects of exercise versus exercise in combination with cognitive training were included in the review.

Primary outcome measures quantifying cardiovascular fitness level, exercise performance, cognition, and aspects of cognitive function were included in the study. Articles were excluded if they did not involve a valid research design or utilize outcome measures for cognition.

### 2.2. Search Methods and Identification of Studies

Studies were identified by searching the CINAHL, Cochrane Wiley Library, Ovid Embase, and Ovid MEDLINE electronic databases from 1995 to the present, with the final search being run in June 2017. The following search terms were employed to search all trials registries and databases: cardiovascular disease*, heart disease*, vascular disease*, stroke, transient ischemic attack, exercise, exercise movement technique, sports, cognition, awareness.

After running the defined search strategy, de-duplication was performed and the remaining article titles and abstracts were independently reviewed by two study investigators (D.A., A.B.) in an unblended, standardized manner to determine whether they met the predetermined eligibility criteria. Each citation was categorized as definitely, possibly, or failing to meet inclusion criteria. For each eligible citation, full-text articles were obtained. For any incongruities in selections, the 3rd investigator (D.H.-R.) was employed for an independent review and resolution. Additional papers were sourced from a scan of the reference list of every article meeting search criteria to ensure a comprehensive search of current literature on the review topic.

Given the inclusion and exclusion criteria applied, researchers found two main categories of studies, those examining the effect of exercise on CVD populations and those on CBVD populations, which were then applied to the data extraction and analysis. See Figure 1 for a schematic of the search process.

### 2.3. Data Extraction

Two researchers independently extracted data from each of the resultant 22 articles and entered the data into a data extraction table for analysis (Appendix A). The extracted data from each study was compared and incongruities were resolved by the third-party reviewer. The data extraction table followed a standardized outline that examined specific attributes of each article including study:research question and purpose,design and method of analysis,sample size and participant characteristics, specifically age, gender, and health condition/diagnosis,inclusion and exclusion criteria,intervention protocols (including type, dose, frequency, and duration) employed,outcome measure(s) utilized,results, including *p*-values,author’s conclusions, andstrengths and limitations.

### 2.4. Critical Appraisal of the Evidence

The PRISMA-P statement and checklist were utilized to ensure a robust preparation and reporting of our protocol for the systematic review [33]. Each component in the data extraction table was reviewed and evaluated for relevance, quality, and findings with respect to our research question. Two researchers then independently rated each study for validity and risk of bias by determining the presence and vigor of randomization and concealment of allocation, blinding of participants and assessors, as well as dropout rates and loss to follow-up. The methodological quality of each paper was assessed using the Oxford Centre for Evidence-Based Medicine (OCEBM) Levels of Evidence criteria, 2011 (http://www.cebm.net/explanation-2011-ocebm-levels-evidence/) and the PEDro Scale, 1999 (https://www.pedro.org.au/english/downloads/pedro-scale/). For the level of evidence and quality rating of each included paper, please refer to Appendix A. Note that findings are reported in the results section below.

## 3. Results

### 3.1. Search Results

The search strategy identified 6824 articles. After screening for duplicates, 5978 papers’ titles and abstracts were reviewed to determine if they met the stated inclusion criteria, resulting in 5890 articles being excluded. The 88 remaining papers underwent independent full-text reads by two of the authors (D.A., A.B.) for a final determination of meeting the inclusion criteria for full review. Ultimately, 22 papers met the inclusion criteria for the systematic review. See Figure 1 for the methods and associated results.

### 3.2. Study Characteristics

#### 3.2.1. Participants

The total number of participants included in the combined 22 trials was 1125, with the largest trial having a sample size of 197 individuals [34]. The majority of the studies were conducted on both males and females; however, there was a far greater number of male participants within each participant pool compared to females (mean = 32.7%). This suggests either a bias of the clinical or research environments or lower referral, uptake, or adherence rates in women with vascular disease [35,36,37]. Only one study was conducted solely with male participants [38], and none with exclusively female participants. Although two studies noted that their participants were Caucasian [34,39], and another cited 9.2% African-American and 2.8% Native American participants [40], the majority of authors did not provide these demographics. In addition, none of the 22 papers cited the socio-demographic details of their participant groups. Interestingly, 10 studies investigating individuals with CVD enrolled participants with HF, while 2 studies included participants diagnosed with either coronary artery disease (CAD), coronary heart disease, hypertension, or documented history of myocardial infarction [41,42]. Seven studies investigating individuals with CBVD included both ischemic and hemorrhagic stroke participants, no longer than 1-year post-stroke, and only three investigations included participants affected by ischemic stroke only [43,44,45]. Across all studies, the participants’ ages ranged from 40 to 85 years, with one study including participants 19 years or older [46]. Additional study participant demographic details can be found in Appendix A.

#### 3.2.2. Outcome Measures

Neuropsychological test batteries were used to assess cognitive function including domains of general cognitive function, attention, executive function, language, and memory. Across all studies examining CVD, the Mini Mental State Examination (MMSE) and Modified Mini Mental State Examination (3MS) were used to assess general cognitive function, while cerebrovascular studies utilized the Montreal Cognitive Assessment (MOCA) and Addenbrooke’s Cognitive Examination—Revised (ACER). Attention and executive function were measured by multiple tools, mainly Trail Making Test A and B, Digit Symbol Coding, Letter Number Sequencing, and the Stroop Color and Word Test. Memory was predominantly assessed using the California Verbal Learning Test—Second Edition (CVLT-II), while language was evaluated with the Boston Naming and Animal Fluency Tests. Physical fitness was primarily assessed using graded exercise tests, the 6-Minute Walk Test (6MWT), and 2-Minute Step Test (2MST). An accelerometer was used as an objective measure of fitness in three studies [47,48,49]. Other objective measures for vascularity included Transcranial Doppler (TCD) to investigate CBF [39,41,44,49], and thoracic electrical bioimpedence for assessment of hemodynamic parameters [49]. El-Tamawy et al. (2014) measured serum brain-derived neurotrophic factor (BNDF), a type of neurotrophin in the brain known to enhance long-term memory, learning, and mental performance and increase the resistance of the brain to insults and degradation with aging. One study used MRI to measure muscle cross-sectional area (CSA) [50]. For further details on outcome measures see Appendix A.

### 3.3. Interventions

Caspersen et al. (1985) were the first to define and distinguish between the terms ‘exercise’, ‘physical activity’, and ‘physical fitness’, terms which describe different concepts and, yet, are often interchangeable. They wrote: “Physical activity is defined as any bodily movement produced by skeletal muscles that results in energy expenditure. The energy expenditure can be measured in kilocalories. Physical activity in daily life can be categorized into occupational, sports, conditioning, household, or other activities. Exercise is a subset of physical activity that is planned, structured, and repetitive, and which has as a final or an intermediate objective of the improvement or maintenance of physical fitness. Physical fitness is a set of attributes that are either health- or skill-related.” [51]. Despite these known and accepted distinctions, many investigators use the terms ‘activity’, ‘physical activity’, and ‘exercise’ interchangeably. Therefore, for our purposes, aerobic exercise refers more broadly to interventions described as a mode targeting primarily the function of the cardiovascular system. Strength training is those interventions focusing primarily on the skeletal muscle system. Both may result in structural, metabolic, and physiological responses. Physical activity, then, is those interventions described as PA but without sufficient detail to be characterized as aerobic or strength training. The characteristics of the included studies are summarized in Appendix A.

Cardiovascular disease studies:

The nature of the interventions employed in the 12 studies examining individuals with known CVD included in our literature review can be broadly grouped into two intervention categories: (1) aerobic exercise training and (2) physical activity.

(1) Aerobic Exercise Training: Five studies examined the effects of an aerobic exercise intervention on memory and cognition in individuals with CVD. Across all studies, aerobic exercise was conducted within a CR program [38,41,49,52,53]. The frequency of exercise sessions varied widely and ranged from 2 to 5 times per week, with the majority of studies prescribing 2 to 3 times per week. The intensity of exercise ranged from moderate to high, at 60–70% heart rate maximum (HRmax) [52] and 70–80% HRmax, while the remaining studies specified intensity as ‘customized’ based upon performance in an exercise stress test [41,52,53]. The duration of exercise sessions ranged from 50 to 75 min, with majority of studies prescribing 50–60 min sessions. Modes of aerobic training included rowing ergometry, walking/treadmill walking, stationary bicycling, elliptical, arm exercises (not specified), and stepping or stair machine. Only one study did not specify the type of aerobic exercise [52]. For further information on intervention protocols, see Table 1.

The longer duration studies investigated between 12- and 18-week long CR interventions and demonstrated improvement in global cognition, attention, executive function, and memory following aerobic exercise training [41,49,52,53]. Community-dwelling older individuals with HF (NYHA Class II–IV) were included in three of the trials [41,49,53], while older adults with coronary heart disease were included in one study [40]. Stanek et al. (2011) and Alosco et al. (2014) conducted comparable 12-week CR interventions and found that 60-minute aerobic circuit training sessions, 3 times per week, not only improved participant’s cardiovascular fitness but additionally demonstrated significant improvements in global cognition, attention, and psychomotor speed [41,53]. Moreover, Alosco et al. (2014) found that participants retained these improvements in cognition when re-tested 12 months later, while Stanek et al. (2011) found that vascular measures, specifically CBF, increased in the anterior cerebral artery (ACA) following aerobic exercise training [41,53]. In another trial, Gunstad et al. (2005) examined the effects of 75-minute exercise sessions consisting of 15 min of intervals and 45 min of continuous aerobic exercise, 3 times per week for 12 weeks, reporting improvements in attention and executive functions, specifically complex attention and psychomotor speed [52]. In a slightly longer, 18-week CR program, Tanne et al. (2005) reported 45 min sessions of continuous aerobic exercise, twice a week, significantly improved attention and psychomotor speed. However, CBF in the MCA and cerebrovascular reactivity remained unchanged compared to controls [49].

A short-duration (14 days) exercise intervention was investigated by Carles et al. (2007) [38]. Older males with heart failure significantly improved attention, executive function, and memory following 50 minutes of aerobic exercise at 70–80% HRR, 5 times per week. Interestingly, Carles et al. (2007) also examined the effect of brief bouts of exercise on cognition and found that participants’ performance on tracking tasks improved when completed during exercise compared to at rest, while cognitive solicitation tasks only improved following 2 weeks of aerobic exercise training [38]. Together, these findings suggest that both brief bouts of exercise and aerobic exercise training can produce cognitive benefits and that exercise training may improve the effect of acute exercise on cognitive function.

Across all trials, participants improved cardiovascular fitness compared to baseline, and statistical analysis showed positive associations between greater fitness levels and improved performance in cognitive tests in older individuals with known CVD [38,41,49,52,53]. This suggests that higher levels of fitness in older individuals with CVD may be associated with improved cognitive functioning. The results of these five studies are in agreement, suggesting that CR programs of both short and long duration produce benefits that extend beyond improvements in cardiovascular fitness and benefit cognitive function, specifically global cognition, attention, and executive function [38,41,49,52,53]. Benefits were demonstrated in both continuous [49,52] and interval [41] aerobic exercise training and were noted in brief bouts of exercise [38]. However, the quality and generalizability of these results are limited due to the small sample size and lack of randomization across trials [38,41,49,52,53]. None of the five trials were RCTs but rather cross-sectional case studies. Only one study included a control group [49], one study included male participants only [38], and only one study included alternate versions of outcome measures of cognition [52] to reduce the risk of practice effect. More importantly, specific parameters for intensity were only described in two of the studies [38,49], preventing any correlations being made between the effect of aerobic exercise intensity on cognition. Ultimately, longitudinal RCTs utilizing exercise interventions must be conducted in order to better understand the relationship between exercise and cognition in those with CVD, and improve current knowledge on the effect of different exercise parameters (duration, intensity, frequency) and cognition.

(2) Physical Activity: Seven studies explored the relationship between physical activity and cognition in older individuals with known CVD [34,39,42,47,48,53,54]. In six of the studies, physical fitness and cognitive function were measured in older individuals with HF [34,39,42,47,48,54], while one trial included both HF and CAD patients [42].

Taken together, the key findings obtained from these seven trials suggest an association between decreased cardiovascular fitness and reduced cognitive function in older individuals with CVD. Alosco et al. (2014) reported that participants who demonstrated poorer performance on the 2MST showed decreased performance in cognitive domains of executive function, global cognition, and language [53]. Similar findings were demonstrated by Alosco et al. (2015), who additionally found that decreased cognition and reduced cardiovascular fitness was negatively associated with ADL function in older men and women with HF [34]. Baldasseroni et al. (2010) reported that individuals who walked shorter distances on the 6MWT also had reduced cognitive function, with the lowest 6MWT scores having the worst performance in cognition tests [42]. Furthermore, Garcia et al. (2013) examined the association between metabolic equivalents of task (METS) from a graded exercise test and cognitive function, reporting higher fitness levels were associated with better cognitive function, specifically in attention, executive function, and memory [54]. When looking at participants’ 2MST performance, these authors also found that participants with poorer 2MST performance had worse performance on attention and executive function tasks, while no association was found when using the Duke Activity Score Index (DASI), highlighting the efficacy of the 2MST as a valid and reliable outcome measure for identifying patients with cognitive impairment. The results of these four trials suggest a possible association between cardiovascular fitness levels and cognitive function in older adults with CVD [34,42,53,54].

Physical activity level was measured by an accelerometer over 7 days in the remaining three studies [39,47,48], which found that lower levels of daily PA were associated with poorer cognitive function in older adults with CVD. Across all studies, individuals with CVD had lower levels of daily PA and spent more time in sedentary activities and intensities. Fulcher et al. (2014) reported that participants with reduced daily step count and decreased total minutes spent in moderate-to-vigorous activity demonstrated poorer performance in attention, executive function, global cognition, and processing speed compared to those with higher values [47]. Alosco et al. (2015) found that when re-tested 12 weeks following accelerometer wear, participants had increased sedentary activity and reduced overall PA, which independently predicted declines in attention and executive function [48]. Similarly, Alosco et al. (2014) found that lower daily step count and less time spent in moderate intensity activity not only predicted poorer attention and executive function, but also predicted poorer CBF of the MCA, at 12-months follow-up [39]. The results of these three trials suggest that increased sedentary activity is negatively correlated to cognitive functioning [39,47,48].

Across all seven trials, reduced cardiovascular fitness was associated with poorer cognitive function, specifically within attention, executive function, and global cognition, in older adults with CVD [34,39,42,47,48,53,54]. Again, this suggests that higher levels of fitness in older individuals with CVD may be associated with improved cognitive functioning. However, the quality of these studies is poor due the single period of cross-sectional data collection and the case study design. These studies only suggest a possible correlation, not causation, between physical fitness and cognitive function in older adults with CVD. Additionally, the generalizability of all trials is poor, due to small sample sizes [34,39,42,47,48,53,54]. Further investigations would be pertinent in determining whether a supervised exercise program aimed at improving cardiovascular fitness could have greater effects on cognition in older individuals with known CVD.

Cerebrovascular disease studies:

The nature of the interventions from the 10 studies examining CBVD included in our literature review can be broadly grouped into three categories: (1) aerobic exercise training, (2) resistance training, and (3) combined aerobic and resistance training.

(1) Aerobic Exercise Training: In total, five investigations examined the effects of an aerobic exercise intervention on memory and cognition in individuals with CBVD against a similar control group [43,44,45,55,59]. The frequency of exercise sessions ranged from 2 to 3 times per week, while intensity varied widely and ranged from moderate to high at 40–80% HRR, with the majority of studies utilizing 70% HRmax. However, two studies specified intensity as ‘intensive’ at moderate to high values but did not specify any intensity percentage [41,42]. The duration of exercise sessions ranged from 30 to 60 min, with most sessions consisting of 45 min of aerobic activity [43,44,45]. Modes of aerobic training included walking/treadmill walking and stationary bicycle, with one study [59] not describing the mode of aerobic exercise employed.

The shorter duration studies investigated 8 weeks of aerobic exercise training and reported significant improvements in cognitive function in older individuals with CBVD [43,44,45,55]. El-Tamawy et al. (2012) and El-Tamawy et al. (2014) conducted similar aerobic exercise interventions in post-ischemic stroke participants and found that 45-minute of continuous exercise on a stationary bicycle, 3 times per week, resulted in not only improved attention, memory, and visuospatial abilities, but also increased serum BDNF levels [43] and CBF in the MCA [44]. These findings suggest positive correlations between both increased MCA flow velocity and higher serum BDNF levels with cognitive function. Quaney et al. (2009) examined the effect of 45-minute aerobic exercise sessions, 3 times per week at 70% HRR, and reported improvements in cognitive function, ambulation, balance, and transfer speed. Despite only finding modest improvements in fitness, increases in information processing speed and attention were found following the exercise intervention [45]. In a more recent study, Blanchet et al. (2016) demonstrated improvements in attention following 30-minute sessions of aerobic exercise at 60–70% HRmax, twice a week, that were retained at 3-months follow up [55]. Although this trial found no improvement in fitness or episodic memory function after the intervention, levels did not decline from baseline when measured immediately after the 8-week intervention and at 3-months follow up [55]. Taken together, the findings of these four studies highlight the efficacy of aerobic exercise intervention on improving various domains of cognitive function in older individuals with CBVD, even over a short duration of time. The results of these studies are not without limitations. Two trials failed to include a control group [43,44], preventing completion of separate group analysis. Across all studies, a small sample size was used, making it difficult to detect true differences in intervention groups [43,44,45,55]. More importantly, two trials did not specify aerobic exercise intensity but rather described it as ‘intensive’ [43,44], limiting the ability to examine an effect of different intensities of aerobic exercise on cognitive function.

A longer-duration (6-month) aerobic exercise intervention was investigated by Tang et al. (2016). Interestingly, the 60-minute exercise sessions performed at 40–80% HRR, 3 times per week, did not produce improvements in cognitive function in older adults 1-year post-stroke [59]. These findings do not align with those of the previous four studies; however, the quality of these results are poor, due to the absence of a specified exercise program within the intervention. The authors attributed this to the study being a secondary analysis of another RCT examining the effect of high- versus low-intensity training on cardiovascular outcomes, rather than cognition. As a result, the exercise protocol itself, or the outcomes selected in the study, may not have been powerful enough to produce and detect changes in cognitive function [59]. Nonetheless, this study outlines how the current understanding of the effect of aerobic exercise on cognitive function remains unclear and that further research needs to be conducted to further explore and strengthen the understanding of this relationship.

(2) Resistance Training: A single study [50] explored the effects of a 12-week resistance exercise training intervention on memory and cognition in older adults with CBVD. Their study included 32 participants, with 16 each in the control group and the intervention group. Fernandez-Gonzalo et al. (2016) found that four sets of 7-repetitions maximum (7RM) on a unilateral eccentric-overload flywheel leg press, twice a week, produced not only improvements in attention, executive function, working memory, and speed of information processing, but also increased muscle CSA and power output of the quadriceps muscle in the affected limb [50]. This is the only study included in this review to examine the effect of resistance training alone, yet the findings provide evidence that cognitive function may improve with high intensity, low repetition strengthening exercises. Of note is that the small sample size limits the generalizability of the results.

(3) Combined Training: The remaining four studies examining exercise and cognition in older adults with CBVD investigated multicomponent interventions [46,56,57,58]. The frequency of exercise sessions ranged from 2 to 5 times per week, with majority of studies prescribing exercise 2 to 3 times per week. Aerobic exercise intensities ranged widely from 50% peak oxygen uptake [56] to 40–70% HRR [60]. One trial did not specify intensity used [46], while another trial described intensity as ‘moderate’ [56]. Resistance exercise intensities were only described in the intervention of Marzolini et al. (2013), ranging from 50–60% 1RM on the unaffected limb and >50% 1RM on the affected limb [57]. Two trials utilized the Fitness and Mobility Exercise (FAME) program protocol for resistance training [46,58] while one used resistance bands to determine intensity [56]. Duration of exercise sessions consisted of 30–60 min, with majority of studies prescribing 60 min of exercise training [46,57,58]. The characteristics of these studies are summarized in Table 2.

Long term interventions were investigated in all four studies, ranging from 12 weeks to 9 months duration, and significant improvements in both cognitive function and functional capacity were reported across the studies [46,56,57,58]. Kluding et al. (2011) found that 30-minute sessions of aerobic exercise at 50% peak oxygen uptake combined with 1 set of 10 repetitions of lower extremity strengthening with resistance bands over 12 weeks, produced improvements in working memory, attention, and skeletal muscle function [56]. In a similar study over 6 months, Marzolini et al. (2013) found improvements in general cognition, attention, executive function, concentration, visuospatial function, and fat-free mass following 60-minute of aerobic exercise at 40–70%HRR, 5 times per week and 10–15 repetitions of strength training at >50% 1RM, 1 to 2 times per week [57]. Across both studies, improvements in cognition were reported despite the duration or intensity of aerobic and resistance exercise and small bouts of strength training were sufficient in producing functional muscle changes [57,58]. Rand et al. (2010) also found that 60 minutes of moderate intensity aerobic and resistance training twice a week, in addition to weekly sessions of recreational and leisure activities, resulted in improved executive function and memory [58]. In a slightly longer intervention (9 months), Lui-Ambrose et al. (2015) found that 60 minutes of community-based exercise training, including aerobic, resistance, and balance exercises twice a week, and 1 hour of recreational activities weekly, produced increases in set shifting, selective attention, conflict resolution, and working memory [46]. Across all four trials, a positive association was seen between improvements in functional capacity and increases in cognitive function following combined aerobic and resistance training in older adults with CBVD [46,56,57,58]. These findings are in agreement with one another; however, the generalizability is low due to small sample sizes and a heterogeneous sample of participants, limiting the ability to detect difference between groups [46,56,57,58]. Further studies utilizing large sample sizes, longitudinal RCTs must be conducted to strengthen current knowledge on the relationship of the effect of various types of exercise on cognitive function in older adults with CBVD.

### 3.4. Study Quality

The Oxford Centre for Evidence-Based Medicine Levels of Evidence (March, 2011; http://www.cebm.net/explanation-2011-ocebm-levels-evidence/) and the PEDro Scale, 1999 (https://www.pedro.org.au/english/downloads/pedro-scale/) were utilized to determine the levels of evidence of the included studies. Due to the absence of systematic reviews within our study, the highest score given was a 1B and an 8 using the PEDro scale. Eight studies were rated 1b, 10 trials obtained a rating of 2b, and the remaining four trials were given a rating of 3b. Fourteen trials were given PEDro 5, three studies obtained a 6, six studies were given PEDro 8, and the remaining study obtained a 7. For more information on quality rating for each trial, see Table 2.

## 4. Discussion

To our knowledge, this is the first systematic review exploring exercise as an intervention to improve cognition, including memory functions, in patients with a diagnosis of CVD or CBVD. Given the evidence to suggest a direct relationship between vascular disease and declines in cognition, exploring the effects of this non-invasive and cost-effective intervention is potentially of great significance. Due to an anticipated scarcity of research, we also chose to include cross-sectional research looking at PA and other measures of fitness. In addition, we decided to include aerobic interventions within the population of individuals post-stroke, as stroke is an indicator condition for CBVD. Collectively, the evidence suggests that aerobic exercise can increase multiple cognitive functions, specifically within areas of global cognition, attention, executive function, memory, and information processing speeds. Furthermore, there appears to be greater evidence for improvements in executive functions and fewer studies finding changes to memory functions following aerobic exercise. Some research has also found that these improvements in cognitive performance have been related to known physiological changes.

The majority of the studies retrieved by our search implemented aerobic training as the primary intervention, which was effective in increasing both memory and executive functions. One resistance training study was included in this review [50]. Authors found that resistance training with low volume, high load exercises could increase attention, executive function, working memory, and speed processing, along with increases in strength and muscle CSA. Others have found that loads of 50–80% 1RM are required to see significant cognitive effects, with greater improvements using higher loads closer to 80% 1RM [57]. Resistance training has been proven to be an effective method of increasing neurocognitive performance with additional increases in brain matter [32,61,62]. However, combined interventions have been previously shown to be more effective than just aerobic or resistance training on their own [30,60]. Furthermore, a meta-analysis found that, among individuals with CAD, combined training can achieve larger increases in aerobic capacity when compared to aerobic training alone [33]. Unfortunately, due to the disparate research methodologies of these studies, we were unable to make such statistical comparisons. Nevertheless, amongst the four studies that implemented combined interventions, there were marked increases in both executive function and memory. We also noted that the exercise parameters prescribed to participants, within combined training, appeared similar or less than either aerobic or resistance training on its own. In addition, some combination studies also managed to include recreational activities and balance training. Finally, Hasan et al. (2016) reviewed and synthesized the data from clinical and animal studies in an attempt to determine optimal aerobic training parameters to enhance cognitive outcomes post-stroke. Although investigating aerobic interventions, they too found that, in the studies with human participants, combined training protocols more consistently improved cognition [63].

Although most studies demonstrated that exercise is correlated with cognitive improvement, some have failed to show these trends. While there could be various reasons for this outcome, intervention duration is commonly discussed. Studies within our review range from 2 weeks to 9 months, with varying levels of success. Prior literature has compared study length and determined that there is no added increase in cognitive performance when comparing short-duration (1–3 months) to medium-duration (4–6 months) interventions; however, long-duration training programs of more than 6 months showed the greatest benefits [29]. This is further supported by the meta-analysis by Marzolini et al. (2012) in which greater improvements in executive function was observed in longer-duration studies [32]. However, our review found that exercise interventions implemented for fewer than 6 months have been able to show improvements in cognitive, including memory, functions.

Among the studies reviewed, the frequency of 2 to 3 times per week was commonly selected. However, we retrieved a study that found cognitive and memory enhancement within a 2-week aerobic intervention, when exercising 5 times per week [38]. Other researchers have investigated how exercise frequency can influence cognitive improvements [64]. Using aerobic exercise, three groups were used to explore this question, one was exercising 3–4 days per week, another 5–7 days per week, and a control group [64]. Those that participated in more frequent aerobic exercise 5–7 × /week saw larger improvements in cognitive function (reaction time, attention, and cognitive flexibility) over those who exercised 3–4 times per week. Researchers concluded that a dose–response relationship exists between exercise and improvements in cognitive function [64]. Therefore, we suspect that exercise frequency is another parameter that could have altered results obtained. Although we can extrapolate from the collective data demonstrating frequency as a positive factor in the context of consistent aerobic exercise prescription durations, that the volume of exercise (calculated as frequency times duration) has a positive effect on cognition, this is not explicitly addressed in the studies reviewed.

Lastly, exercise intensity is a potential variable which may be key to obtaining optimal cognitive improvements. Previously, the relationship between exercise intensity and cognition has been hypothesized to be an inverted-U, where moderate intensity achieves the best results [65,66]. However, Tang et al. (2016) was unable to achieve cognitive improvements at the end of the 6-month intervention, while participants were working at 58.1% HRR [59], suggesting that a moderate level of intensity may not be sufficient, in those with vascular disease, to obtain significant improvements in cognition. Intensity levels have also been investigated in populations of individuals with MCIs [67]. It was found that among MCIs patients, exercising at 40% or 60% HRR was not sufficient to improve cognitive function [68]. Authors concluded that low or moderate intensity was not sufficient to see any cognitive changes post-aerobic exercise training. Conversely, recent studies have been able to achieve improvements in selective attention and short-term memory following an acute bout of high intensity interval training [66]. These findings suggest that perhaps intensity is central to the outcomes but that it may be linear rather than following an inverted-U model. Studies included in this review were able to attain significant global cognitive improvements using moderate to high intensities.

As previously mentioned, CBF reduction is seen with aging and lower CBF has been correlated to lower brain function [22,23]. In addition, several studies have seen correlations between the volume of PA and aerobic fitness to CBF [53,69]. Furthermore, chronic HF patients (NYHA class III and IV) show approximately 30% reduction in CBF, compared to age-matched controls, which can be completely reversed post-heart transplant [68]. Therefore, it was theorized that aerobic exercise promotes cognitive function via increases in CBF [29,56,70]. Three of the studies retrieved investigated changes in CBF and in brain function [41,43,49]. Although all studies reported significant cognitive improvements, post-aerobic exercise intervention, not all studies were able to attain significant increases in CBF. Thus, we are unable to support or deny the relationship of CBF and cognitive function within those that suffer from vascular disease. Further research is required to understand how CBF responds to aerobic exercise and cognition.

Increasingly, BDNF is attributed to playing a role in increasing the brain’s resistance to damage and degeneration with aging, with higher BDNF indicating better brain well-being [71]. Reviews of the current literature had found that BDNF is increased following both acute and chronic aerobic interventions [71,72]. Other research has also been able to show that those who are more physically active have elevated resting BDNF [68,71]. Furthermore, cognitive improvement following an aerobic intervention has been related to increases in BDNF, which is dependent on exercise intensity [71,72,73,74]. Our search retrieved one study that used BDNF as an outcome measure [43]. Their findings add to current literature, showing that patients with vascular disease, post-aerobic exercise, also see increases in BDNF and that these increases are associated to cognitive improvement [43]. BDNF is another potential mechanism in which greater cognitive function is achieved; however, more research is needed to determine the direct effects among persons with CVD.

### Limitations and Implications

We have collected studies that have used exercise interventions to improve cognitive function and systematically compared study protocols and findings. However, a limitation of this review is lack of statistical analysis to determine which of the above-mentioned parameters have the greatest potential to attain clinically significant changes in cognition, including memory. Furthermore, a majority of patients used within the selected studies were heart failure and stroke patients. As a result, the current findings may not be generalizable to all CVD patients. Lastly, the study quality of studies used within this review have been rated relatively low (The Oxford Centre for Evidence-Based Medicine Levels of Evidence and the PEDro Scale). There were multiple factors leading to the lower quality ratings, including, but not limited to, participant numbers. Of note was the absence of studies focused on female populations and those of non-Caucasian ethnicity. In addition, studies failed to consistently report data (e.g., exercise protocol) that would help understand results.

## 5. Conclusions

The current study reveals evidence that aerobic exercise is able to positively affect both memory and executive functions in individuals with vascular disease. Collectively, there is evidence of a dose–response relationship, with trends towards greater volume resulting in further cognitive outcomes. We have discussed various parameters which could be manipulated to change the degree of impact; however, at this time we are unable to recommend specific exercise prescription parameters for those with known vascular disease. Although there have been some studies investigating potential physiological mechanisms by which aerobic exercise can improve brain functions (CBF, BDNF, hypertension, CO), both exercise prescription and the responses to exercise are complex processes with more prospective, controlled trials required to determine the relationship between them. Future research should include female participants, psychological variables which may affect outcomes, and MRI data to better explain how exercise improves cognitive function.

## Figures and Tables

**Figure 1 jcm-08-00294-f001:**
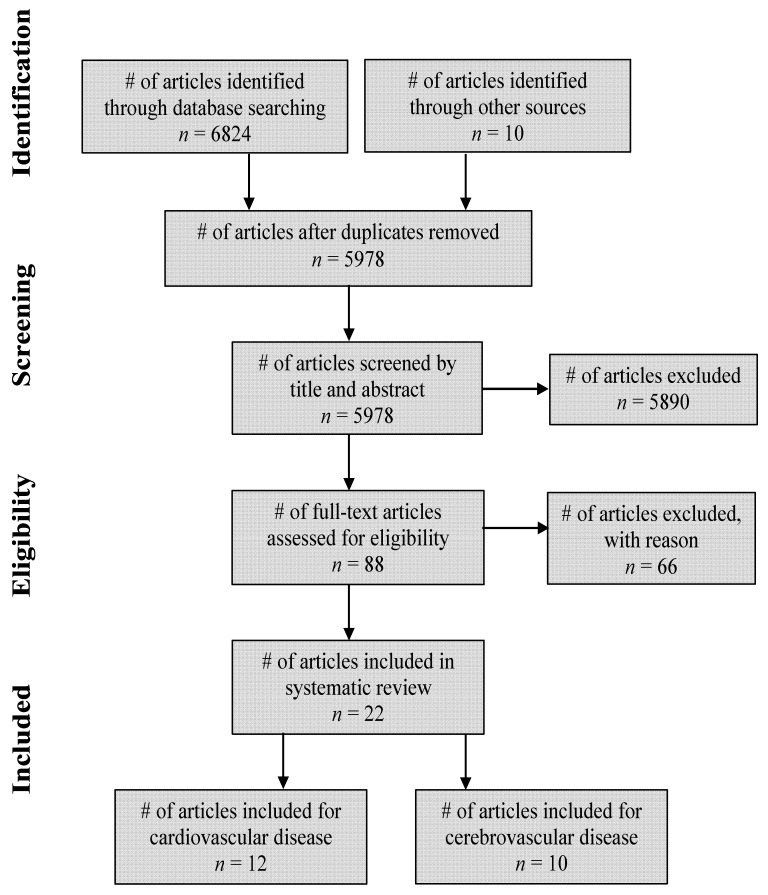
Search process and results.

**Table 1 jcm-08-00294-t001:** Study Outcome Measures.

Study (year)	Cognition Outcome Measures	Physical Fitness Outcome Measures	Additional Outcome Measures
**Cardiovascular Studies**			
Alosco et al. (Feb 2015) [34]	**Attention/Executive Function** Trail Making Test A/B LNS DSCT **Memory** CVLT-II	2MST	Lawton–Brody ADL scale
Alosco et al. (April 2015) [48]	**Attention/Executive Function** FAB LNS DSCT **Memory** CVLT-II **Language** BNT Animal Fluency Test	Accelerometer × 7 days	BDI-II
Alosco et al. (April 2014) [39]	**Global Cognition** 3MS **Attention/Executive Function** Trail Making Test A/B FAB **Memory** CVLT-II	Accelerometer × 7 days 2MST	TCDBDI-II
Alosco et al. (Aug 2014) [53]	**Global Cognition** MMSE **Attention/Executive Function** Trail Making Test A/B DSCT **Memory** CVLT-II **Language** BNT Animal Fluency Test	2MST	N/A
Alosco et al. (2012) [40]	**Global Cognition** 3MS **Attention/Executive Function** SCWT FAB LNS DSCT **Memory** CVLT-II **Language** BNT Animal Fluency Test	2MST	BDI-II
Baldasseroni et al. (2010) [42]	**Global Cognition** MMSE	6MWT	Sickness Impact Profile (Psychosocial subscale) HR-QOL ADL Scale
Carles et al. (2007) [38]	**Attention** Trail Making Test Cognitive solicitation **Motor Precision** Tracking Task	Incremental Exercise Test	N/A
Fulcher et al. (2014) [47]	**Global Cognition** 3MS **Attention/Executive Function** Trail Making Test A/B SCWT LNS **Memory** CVLT-II	Accelerometer × 7 days	BDI-II
Garcia et al. (2013) [54]	**Global Cognition** 3MS **Attention** Trail Making Test A/B SCWT FAB LNS DSCT **Memory** CVLT-II **Language** BNT Animal Naming Test	GXT 2MST DASI	BDI-II
Gunstad et al. (2005) [52]	**Psychomotor Speed** Trail Making Test A Grooved Pegboard DSCT **Language** Animal Category Fluency	GXT	BDI-II
Stanek et al. (2011) [41]	**Global Cognition** MMSE **Attention/Executive Function** Trail Making Test A/B Grooved Pegboard FAB LNS **Memory** HVLT-R Delayed Recall and Recognition BVMT-R **Language** BNT Animal Naming Test	GXT Modified ramp protocol	TCD
Tanne et al. (2005) [49]	**Global Cognition** MMSE **Attention** Trail Making Test A/B SCWT Continuous Performance Test **Memory** Rey–Osterrieth Complex Figure **Phenomic/Semantic Fluency** Verbal Fluency Tests	6MWT GXT Modified Bruce Protocol	Thoracic Electrical Bioimpedence TCD Breath-Holding Index
**Cerebrovascular Studies**			
Blanchet et al. (2016) [55]	**Memory** HVLT-R **Attention** CPT Brown–Petersen paradigm	GXT recumbent cycle ergometer	N/A
El-Tamawy et al. (2014) [43]	**Global Cognition** ACER	N/A	Serum BDNF
El-Tamawy et al. (2012) [44]	**Global Cognition** ACER	N/A	TCD
Fernandez-Gonzalo et al. (2016) [50]	**Attention/Executive Function** SCWT DSCT CPT-II **Memory** RAVLT	Peak power Muscle CSA (MRI)	BERG balance scale
Kluding et al. (2011) [56]	**Attention** Flanker Task Digit Span Backward Test **Memory** SIS	6MWT	Fugl–Meyer Test
Lui-Ambrose et al. (2015) [46]	**Global Cognition** MOCA **Attention/Executive Function** Trail Making Test A/B SCWT Digit Span Forward and Backward Test	6MWT	BERG balance scale
Marzolini et al. (2013) [57]	**Global Cognition** MOCA	GXT cycle ergometer ATGE 1RM 6MWT	CES-D
Quaney et al. (2009) [45]	**Attention** Trail Making Test A/B SCWT WCST SRTT **Conditional Learning** PGFM	GXT	Fugl–Meyer Test
Rand et al. (2010) [58]	**Attention/Executive Function** Trail Making Test B SCWT Digit Span Backward Test Walk While Talking Task **Memory** RAVLT	6MWT Dynamometer	Geriatric Depression Scale
Tang et al. (2016) [59]	**Global Cognition** MOCA **Attention/Executive Function** Trail Making Test B SCWT DSCT	GXT cycle ergometer6MWT	CES-D Scale

1RM, 1 Repetition Maximum; 2MST, 2-min step test; 3MS, Modified Mini Mental State Examination; 6MWT, 6-min walk test; ACER, Addenbrooke’s Cognitive Evaluation; ADL, Activities of Daily Living; ATGE, Anabolic Threshold Gas Exchange; BDI-II, Beck Depression Inventory II; BDNF, Brain derived neurotrophic factor; BNT, Boston Naming Test; BVMT-R, Brief Visual Memory Test; CES-D, Centre for Epidemiologic Studies Depression Scale; CPT-II, Continuous Performance Test; CSA, Cross-Sectional Area; CVLT-II, California Verbal Learning Test—Revised; DASI, Duke Activity Status Index; DSCT, Digit Symbol Coding Test; FAB, Frontal Assessment Battery; GXT, Graded Exercise Test; HR-QOL, Health Related Quality of Life Scale; HVLT-R, Hopkins Verbal Learning Test—Revised; LNS, Letter Number Sequencing; MMSE, Mini Mental State Examination; MOCA, Montreal Cognitive Assessment; N/A, not applicable; PGFM, Predictive Grip Force Modulation; RAVLT, Rey Auditory Verbal Learning Test; SCWT, Stroop Color and Word Test; SIS, Stroke Impact Scale; SRTT, Serial Reaction Timed Task; TCD, Transcranial Doppler; WCST, Wisconsin Card Sorting Task.

**Table 2 jcm-08-00294-t002:** Intervention Characteristics.

Study (year) Oxford Rating * PEDro Rating	Intervention	Outcome Measure Timing	Duration (weeks) Frequency (times per week)	Aerobic Training (Duration and Intensity)	Resistance Training (Sets/Repetitions No. of Muscle Groups %1RM)
**Cardiovascular Studies**					
Alosco et al. (Feb 2015) [34] 2B PEDro 5	PF	Single assessment	Single time measurement	2MST	N/A
Alosco et al. (April 2015) [48] 2B PEDro 5	PF	Pre- & post-intervention	12 wk; with 1wk accelerometer	Accelerometer × 7 days	N/A
Alosco et al. (April 2014) [39] 1B PEDro 5	PF	Single assessment TCD measured at baseline and 12-months follow up	1wk	Accelerometer × 7 days	N/A
Alosco et al. (Aug 2014) [53] 2B PEDro 5	AT	Pre- and post-intervention 12-months follow-up	12 wk 3 × /wk	CR; 60 min total Warm up, 40 min circuit training, cool down; Customized intensity + 30 min education	N/A
Alosco et al. (2012) [40] 2B PEDro 5	PF	Single assessment	Single time measurement	2MST	N/A
Baldasseroni et al. (2010) [42] 2B PEDro 5	PF	Single assessment	Single time measurement	6MWT	N/A
Carles et al. (2007) [38] 2B PEDro 5	AT	Pre- and post-intervention	2 wks 5 × /wk	CR; 50 min total Warm up, aerobic exercise, cool down; 70–80% HRR	N/A
Fulcher et al. (2014) [47] 3B PEDro 5	PF	Single assessment	1 wk	Accelerometer × 7 days	N/A
Garcia et al. (2013) [54] 3B PEDro 5	PF	Single assessment	Single time measurement	GXT TM: elevation increase every min, speed increased every 3 min to increase workload by 15%	N/A
Gunstad et al. (2005) [52] 1B PEDro 5	AT	Pre- and post-intervention	12 wks 3 × /wk +2 × /wk education	CR; 75 min total Warm up, 15 min intervals, 45 min continuous exercise; Intensity not specified	N/A
Stanek et al. (2011) [41] 2B PEDro 5	AT	Pre- and post-intervention	12 wks 3 × /wk	CR; 60 min total Warm up, 40 min circuit training, cool down; Customized intensity + 30 min education	N/A
Tanne et al. (2005) [49] 1B PEDro 6	AT	Pre- and post-intervention	18 wks 2 × /wk	CR; 50 min total Warm up, aerobic exercise, cool down; 60–70% HRmax	N/A
**Cerebrovascular Studies**					
Blanchet et al. (2016) [55] 1B PEDro 7	AT	Pre- and post-intervention 3-months follow up	8 wks 2 × /wk	30 min plus warm up and cool down; 60–70% HRR Increased 5–10% in last 4 weeks BORG 6–7	N/A
El-Tamawy et al. (2014) [43] 2B PEDro 6	AT	Pre- and post-intervention	8 wks 3 × /wk	75 min total 30 min PT session; stretching, balance, functional training; 45 min aerobic exercise; ‘Intensive’ intensity	N/A
El-Tamawy et al. (2012) [44] 2B PEDro 6	AT	Pre- and post-intervention	8 wks 3 × /wk	75 min total 30 min PT session; stretching, balance, functional training; 45 min aerobic exercise; ‘Intensive’ intensity	N/A
Fernandez-Gonzalo et al. (2016) [50] 1B PEDro 8	RT	Pre- and post-intervention	12 wks 2 × /wk	N/A	4 sets of 7 maximal repetitions <2min contractile activity
Kluding et al. (2011) [56] 3B PEDro 5	CT	Pre- and post-intervention	12 wks 3 × /wk	30 min Warm up, aerobic exercise, cool down; 50% peak 0_2_ uptakeRPE 11–14 +1 hr recreation	Seated exercises Elastic resistance bands of varying resistance 1 set × 10 reps
Lui-Ambrose et al. (2015) [46] 1B PEDro 8	CT	Baseline and every 3 months with a 6-month follow-up	9 months 2 × /wk	60 min total Community-based training; +1 hr/wk recreational activities; Intensity not specified	Not specified FAME protocol
Marzolini et al. (2013) [57] 2B PEDro 5	CT	Pre- and post-intervention	6 months 5 × /wk +2 × /wk RT	60 min total 40–70% HRR BORG 11–16	Task-specific exercises Unaffected limb: 50–60% 1RM Affected limb; >50% 1RM or RPE 13–14 10–15 repetitions
Quaney et al. (2009) [45] 1B PEDro 8	AT	Pre- and post-intervention 8-weeksfollow-up	8 wks 3 × /wk	45 min total Warm up and cool down; 70% HRmax	N/A
Rand et al. (2010) [58] 3B PEDro 5	CT	Baseline, halfway, and post-intervention	6 months 2 × /wk	60 min total FAME protocol ‘moderate intensity’ +1 hr/wk recreation	Not specified FAME protocol
Tang et al. (2016) [59] 1B PEDro 8	AT	1 month pre- and post-intervention	6 months 3 × /wk	60 min Individualized exercise program; 40–80% HRR	N/A

* Oxford Rating; (March, 2011; http://www.cebm.net/explanation-2011-ocebm-levels-evidence/); PEDro Scale (1999; https://www.pedro.org.au/english/downloads/pedro-scale/); 1RM, 1 repetition maximum; 2MST, 2-minute step test; 6MWT, 6-minute walk test; AT, Aerobic training; wks, weeks; CR, cardiac rehabilitation; CT, combined aerobic and resistance training; FAME, Fitness and Mobility Exercise program protocol; GXT, graded exercise test; HRmax, Heart rate max; HRR, heart rate reserve; N/A, not applicable; PF, physical fitness; PT, physiotherapy; RPE, Rating of Perceived Exertion; RT, resistance training; TCD, Transcranial Doppler; TM, treadmill; 3 × /wk, 3 times per Week.

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
