# Peer review of "The Effectiveness of Exercise on Cognitive Performance in Individuals with Known Vascular Disease: A Systematic Review"

_jcm, 2019, doi:10.3390/jcm8030294_

Reviewer 1 Report

Thank you for the opportunity to review this original scientific paper of The Effectiveness of Exercise on Cognitive Performance in Individuals with Known Vascular Disease: A Systematic Review. Editorial/ grammar/ sentence structure along with writing fluency revisions are needed throughout the paper.

 Introduction

-          Line 34-35. In Europe, cardiovascular diseases (CVD) are responsible for more than 4 million deaths annually. Please, check publications and data around the globe and start the paragraph with worldwide data. It is well known the effect of CVD when regarding death, morbidity, disability, and cost.

-          Lack of a perspective more integrative. Please try to shorten the introduction. It is too long.

-          Check other factors that may affect cognitive performance in vascular disease. E.g. stress and anxiety are quite common on these patients and might affect cognitive performance.

 Methods:

-          Please, include manuscripts before 1995. It will improve the quality of your manuscript.

-          Figure 1. It seems there is a mistake. 88 – 62 = 26, not 22.

 Results

-          A brief summary of socio-demographic variables should be provided in the text

-          Were language, race/ ethnicity, and level of education information collected from the studies? It would be interesting to know the profile of the subjects included in the studies.

-          245 when you say 32.7, does it refer to the median value? Please specify in the text

-          246. … lower referral, uptake and/or adherence rates in women with vascular disease [35,36] Please, cite the manuscript Supervia M. Cardiac Rehabilitation for Women: A Systematic Review of Barriers and Solutions.

-          247. Was there any study with female exclusively? Please, specify

-          3.3. Interventions. You should explain the differences between exercise, physical activity and physical fitness in the introduction at the same time you justify the words you use for your search.

-          Table 1. Please, specify the type of exercise intervention. When aerobic exercise was the intervention, please specify if continuous or intervallic in the different studies when known.

-          Please, try to make the results section easier to read.

-          Please, in table 2 consider adding a column about the effectiveness of each intervention. In order to see easily if exercise or just increasing PA might have positive results in the different cognitive performances.

 Discussion

-          Please, provide a general interpretation of results in context to prior evidence.

-          Limitations of the population of the study to a generalization of the results should be addressed (low % of women, sample size,...)

-          Lack of critical perspective more integrative. Other factors that may affect to the level of cognitive performance should be addressed in the discussion.

-          Please, re-write the discussion in order to make it more insightful

-          Please, make a comment about the relevance of to approach cognitive performance in CVD patients.

-          Do you think that primary prevention on CV factors should include a cognitive assessment in order to improve the assessment and management of these patients?

-          549. Please, consider adding a comment about the dose of exercise. It’s not just frequency, it’s the total dose (including frequency and duration of each session).

 Conclusion

-          Please, highlight the need for including more women in the studies.

-          Please specify that more studies including other variables like psychological variables, obstructive sleep apnea, pain,… and so on are needed to see the effectiveness of exercise and PA on cognitive performance.

-          Please, consider to cite that to prescribe exercise and PA in this kind of patients, an integral approach and evaluation of each patient are needed.

 Author Response

Response to Reviewer 1: 

Thank you for your review of this paper, The Effectiveness of Exercise on Cognitive Performance in Individuals with Known Vascular Disease: A Systematic Review and the opportunity to respond to your feedback and suggestions for revisions. We have attempted to make revisions where feasible and a rationale for any revisions which have not been made.

The point-by-point response has been uploaded as a pdf file below.

Reviewer 2 Report

I find the manuscript interesting, overall well written and easy to read.

However, I have some minor comments and questions:

In general I would prefer that small numbers are written out in letter instead of using numerals (e.g. better to write four instead of 4). E.g. lines 302, 351, 352, 359, 363 and many other places, especially when referring to number of studies.

  What are the asterisks (*) in the keywords and line 189 referring to?

Also use of asterisk in Table 2 related to the Oxford Rating and corresponding author?

Line 56-57: high and low IMT - which group contain 0.9 mm? In the same paragraph (<0.9mm) (line 57) and equal to or less (line 64) is written in different ways. Should be written in the same way.

Line 64: When were the participants less active? At follow-up?

Line 133: Is "where" a typo? (compare to "were" in line 132).

Line 143: Unsure of the meaning of the word "slate" here?

Abbreviations not explained before using them: VaD (line 113), PA (line 152)

Line 158: Should the research question also include CBVD?

Figure 1 is first mentioned in line 201. Should be presented in section 3 Results instead. The box in the upper right corner in Figure 1 ( # articles identified through other sources n=10) is not referred to in section 1.3 (line 235). Are any of these ten included in the final twenty-two studies?

The term "stroke" (Figure 1) and "cerebrovascular studies" (Table 1) should be written in the same way if they are exprssions of the same studies.

Line 242: Number in the combined trials? Is that the total number of participants in all groups in the twenty-two included trials? 

Line 298: Hrmax or HRmax (line 297) and also mix in Table 2 (Tanne et al.). Be consistent.

Table 1 vs Table 2. Be consistent in the way you refer to the studies, especially those with the same first author and year (Alosco et al. 2014 and 2015 twice)  vs Alosco et al. Feb 2015 and April 2015. 

Explanations of abbreviations in the tables should be ordered systematically. Either in all over alphabetic order or first in study outcome groups (column headings) and then in  alphabetic order. Abbreviation DCST (Table 1, Alosco et al. (2014): should it read DSCT? For Garcia et al. (2013) BDI-!! should read BDI-II. For Quaney et al. (2009): What does WSCT mean? WCST or SCWT? (typo?).

In line 320 and 323: Which study by Alosco et al. (2014) are you referring to?

Several times wrong use of - between number and time. E.g. line 328 should be 12 weeks not 12-weeks. See also line 475, 476, 478, 484, 485.

Line 453: CG=16 and IG=16; I guess this means control group and iintervention group and numbers in each group. Write it up in a better way?

Various ways to write number of times per week. Better be consistent: E.g, lline 410: 2-3 times per week; line 441: 3 times a week; lines 554 and 556 5-7x/week.

Line 492: Lacking a full stop after the references.

Table 2: Same comment to explanation of abbreviations as to Table 1. Also lacking explanation for CT, RT, RPE and FAME?

Lines 527 and 528: 1RM vs Table 2: 1-RM. Be consistent. 

Lines 511-512: It is always hard to tell if the claim to be the first study on a certain topic is true or not.

Line 533: Why were you unable to make such statistical comparisons?   Unfortunately is a weak reason?  

Line 596: Lacking the term "systematically"? 

Line 602: should it read "rated" instead of "rate"? Also, the study quality could have been discussed in the discussion part, not only in the "Limitations and Implications" section?

Line 694 (ref 34) and Line 737 (ref 52) are the same reference (duplicate).

Well  structured supplemetary file.

Author Response

Response to Reviewer 2:

Thank you for your review of this paper, The Effectiveness of Exercise on Cognitive Performance in Individuals with Known Vascular Disease: A Systematic Review and the opportunity to respond to your feedback and suggestions for revisions. We have attempted to make revisions where feasible and a rationale for any revisions which have not been made.

The complete point-by-point response has been uploaded as a pdf file below.
